# Role of PD-L1 in Gut Mucosa Tolerance and Chronic Inflammation

**DOI:** 10.3390/ijms21239165

**Published:** 2020-12-01

**Authors:** Marina Chulkina, Ellen J. Beswick, Irina V. Pinchuk

**Affiliations:** 1Department of Medicine, Penn State Health Milton S. Hershey Medical Center, Hershey, PA 17033, USA; mchulkina@pennstatehealth.psu.edu; 2Department of Internal Medicine, University of Utah School of Medicine, Salt Lake City, UT 84132, USA; ellen.beswick@hsc.utah.edu

**Keywords:** PD-L1, gastrointestinal inflammation, homeostasis, inflammatory bowel disease

## Abstract

The gastrointestinal (GI) mucosa is among the most complex systems in the body. It has a diverse commensal microbiome challenged continuously by food and microbial components while delivering essential nutrients and defending against pathogens. For these reasons, regulatory cells and receptors are likely to play a central role in maintaining the gut mucosal homeostasis. Recent lessons from cancer immunotherapy point out the critical role of the B7 negative co-stimulator PD-L1 in mucosal homeostasis. In this review, we summarize the current knowledge supporting the critical role of PD-L1 in gastrointestinal mucosal tolerance and how abnormalities in its expression and signaling contribute to gut inflammation and cancers. Abnormal expression of PD-L1 and/or the PD-1/PD-L1 signaling pathways have been observed in the pathology of the GI tract. We also discuss the current gap in our knowledge with regards to PD-L1 signaling in the GI tract under homeostasis and pathology. Finally, we summarize the current understanding of how this pathway is currently targeted to develop novel therapeutic approaches.

## 1. Introduction

The intestinal tract is a complicated and well-orchestrated system, comprising different types of cells, including dendritic cells (DC), macrophages, T cells, B cells, innate lymphoid cells (ILCs), as well as epithelial, mesenchymal, and endothelial cells. The tightly regulated interactions between those cells segregate commensal microbes and maintain a balance to ensure tissue health and regeneration. The continuous crosstalk between gut microbiota and the host gastrointestinal (GI) tract supports a well-balanced relationship between gut microbes and the host’s immune system [1]. One of the major processes in intestinal homeostasis, which controls local inflammatory responses in the gut, is mucosal tolerance [2].

Over the last decade, signaling through the B7 negative co-stimulator, PD-L1, has emerged as a key mechanism for the mucosal tolerance in the gut [3]. The engagement of PD-1 on T cells by PD-L1 inhibits the activation and proliferation of effector T cells [4], inducing those producing inflammatory IFN-g and IL-17A cytokines. Furthermore, PD-L1 has been shown to induce regulatory T cells (Tregs) [5,6]. PD-L1 expression is normally upregulated during inflammation to prevent overt tissue damage [3]. The importance of this molecule in regulating immune responses in the intestinal tract and maintaining tolerance was established in animal models using PD-L1 and/or PD-1 knockouts and transgenic mice [7]. Abnormal expression of PD-L1 and/or PD-L1/PD-1 signaling have been observed in Inflammatory Bowel Diseases (IBD) [8,9,10,11,12,13], *Helicobacter pylori* chronic infection [14,15] celiac disease [16], and GI cancers [17,18,19,20]. Studies in preclinical models and clinical data suggest that PD-L1 may serve as a prognostic marker and therapeutic target in several GI chronic inflammatory diseases and cancer. PD-1 blockade has been shown to reinvigorate exhausted T cells, providing enhanced anti-tumor responses [21]. These observations have led to the development of an anti-cancer immune checkpoint therapy targeting PD-L1/PD-1 signaling [5,22]. Importantly, gastrointestinal adverse events following immune checkpoint blockade in cancer patients point out the importance of the baseline expression of these molecules in gut homeostasis [23,24]. At the same time, targeting immune checkpoints in given subgroups of IBD patients with an abnormality in PD-L1 signaling was proposed as a target for the development of better personalized therapeutic approaches [25].

Here, we review the current knowledge supporting the key role of PD-L1 signaling in the maintenance of gut immune homeostasis and the contribution of the abnormality in PD-L1 signaling to GI inflammatory diseases and cancer. We also discuss the current gap in knowledge about PD-L1 signaling in the GI tract. Finally, we summarize how this pathway is currently targeted for the development of novel therapeutic approaches.

## 2. B7 Immunoglobulin Superfamily in Peripheral Immune Tolerance

PD-L1 (a.k.a. B7-H1, CD274) belongs to the B7 family, which includes more than 10 members (B7-1, B7-2, B7-H1, B7-DC, B7-H2, B7-H3, B7-H4, B7-H5, BTNL2, B7-H6, and B7-H7). This family is one of the most characterized immunoglobulin superfamilies (IgSF) [26]. The B7 co-stimulators all share a similar structure of transmembrane or glycosylphosphatidylinositol (GPI)-linked proteins, which are characterized by targeting the N-terminus of the protein extracellularly through IgV and IgC domains related to the variable and constant domains of immunoglobulins [27]. The B7-CD28 family is phylogenetically divided into three groups—group I consisting of B7-1/B7-2/CD28/CTLA4 and B7h/ICOS, group II containing PD-L1/PD-L2/PD-1, and group III including B7-H3 (a.k.a. CD276), B7x (a.k.a.B7-H4 or B7S1), and HHLA2 (a.k.a.B7H7 or B7-H5)/TMIGD2 (IGPR-1/CD28H) [28]. The B7-1/B7-2/CD28/CTLA-4 pathway is important in modulating central immune tolerance, while PD-L1/PD-L2/PD-1, B7-H3, B7x, and HHLA2 are important in peripheral immune regulation [28]. Together with the TCR–MHC complex, B7 and its receptors play a critical role in the regulation of cell proliferation and cytokine secretion [29]. It was discovered that these molecules play a critical role in tolerogenic immune responses in autoimmunity [30], maternal-fetal immunity [31], host versus graft response [32], and intestinal homeostasis [33].

## 3. PD-L1 in Gut Homeostasis

Over the last decade, studies have shown that PD-L1/PD-1 signaling is critical to regulating both innate and adaptive immune responses in gut mucosa under homeostasis. The expression of PD-L1 by non-hematopoietic cells is suggested to regulate self-reactive T cells or B cells and inflammatory responses in the gut and its associated gut-associated lymphoid tissue (GALTs) [5,34]. PD-L1-positive stromal cells were reported to inhibit granzyme B production in CD8^+^ T effector cells in vitro [35]. PD-L1-mediated signaling on the mesenchymal component of the mucosal lamina propria has been shown to suppress IFN-γ and IL-17A producing T helper (Th) cell responses in the colon [12,36,37]. In general, PD-L1 has been shown to regulate the development, maintenance, and function of inducible Foxp3^+^ Treg in vitro and in vivo [6].

The basal level of PD-L1 expression was detected under homeostasis in both the upper and lower gastrointestinal tract. For example, using immunohistochemistry on the paraffin-embedded tissue, Mezache et al. demonstrated a high basal expression of PD-L1 by epithelial cells within the gastric gland [38]. We reported the expression of PD-L1 on both mRNA and protein levels within mucosal lamina propria in the small intestine and colon [11]. In particular, the constitutive expression of PD-L1 has been reported for small intestinal and colonic epithelial cells and suggested to be critical in the epithelial mediated control over proliferation of the CD4^+^ and CD8^+^ T cells [8]. Our group observed that within the colonic mucosa, CD90^+^ mesenchymal cells were the major cells expressing surface PD-L1 in the normal human colon [11], and suggested the role of that molecule in maintaining immune tolerance in the intestinal tract [12,36]. A similar observation was made by us for murine colonic mucosa, where activated mesenchymal cells, known as α-SMA^+^ myofibroblasts, were the major cells expressing PD-L1 [36].

The expression of PD-L1 and its receptor PD-1 was found to be significant in the germinal centers of the Peyer’s Patches (PPs). PD-L1 was found to be abundantly expressed by dendritic cells, macrophages, B cells, and upregulated in plasma cells, while T cells were major expressors of PD-1 [39]. Additionally, PD-L1/PD-1-mediated signaling was suggested to be critical in the regulation of the plasma cell responses. Lack of the PD-L1 receptor, PD-1, in mice leads to an excess of T follicular helper (Tfh) cells with altered phenotypes resulting in a dysregulated selection of IgA precursor cells in the germinal centers of PPs [40]. This signaling pathway was shown to be critical to the maintenance of a healthy microbiome through the regulation of the IgA selection [39,40]. Indeed, PD-1KO mice have a dramatically alteration in microbiota such as increases in *Erysipelotrichaceae*, *Prevotellaceae*, and *Alcaligenaceae*, while *Bifidobacterium* and *Bacteroidaceae* were undetectable. PD-1 deficiency was shown to affect the selection of IgAs in the gut that resulted in the reduced bacteria-binding capacity of IgA [39]. Furthermore, blocking of PD-L1 during activation of T cells with *Staphylococcus aureus* was reported to reduce the induction of Foxp3^+^CD25^+^CD127^low^ T cells [41]. Additionally, while less studied, PD-L1:B7-1-mediated interactions between the non-hematopoietic components of the colonic mucosa and macrophages were suggested to control intestinal inflammatory responses [42]. Recent evidence suggests that PD-L1 also contributes to the regulation of innate lymphoid cell 2 (ILC2), ILC3, and small intestinal lamina propria lymphoid tissue inducer (LTi) cell function [43] Indeed, PD-L1 deficiency on ILC2s disrupts Th2 polarization and cytokine production, leading to delayed worm expulsion during infection with the gastrointestinal helminth *Nippostrongylus brasiliensis* [44].

The critical role of gut mucosal PD-L1 in tolerogenic responses under homeostasis was pointed out by Reiynoso et al. [45]. Using an iFABP-tOVA transgenic mouse model, in which OVA was expressed as a self-Ag throughout the small intestine and an adoptive transfer of naive OVA-specific CD8^+^ T cells, it was demonstrated that abolishing PD-L1/PD-1 signaling resulted in a break of intestinal tolerance to intestinal self-Ag and induced CD8^+^ T cell-mediated autoimmune enteritis [45].

Despite advances in our understanding of the contribution of PD-L1 to gut mucosal tolerance, significant gaps remain in our knowledge of how this molecule’s expression is regulated during gut homeostasis. We previously reported that signaling through MyD88-dependent TLRs is required to maintain PD-L1 expression on mesenchymal stromal cells in the normal colonic mucosa [36]. The expression of PD-L1 is upregulated by several inflammatory cytokines and growth factors. Indeed, an increase in PD-L1 by IFN-γ was reported in several studies on macrophages, dendritic cells, and lymphocytes [46,47,48,49] as well as on mesenchymal and epithelial cells [8,11]. Importantly, this cytokine was earlier reported to be produced by lymphocytes under homeostasis, in particular in duodenum [50]. TGF-β, which plays a critical role in the colonic mucosal tolerance [51], has also been suggested to be important in the expression of PD-L1 by DCs in the colon [52]. Thus, while further studies are needed to understand the mechanisms contributing to the regulation of PD-L1 expression and its interaction with other positive and negative B7 co-stimulators, it is clear that the maintenance of the basal level of PD-L1 is critical to gut homeostasis.

## 4. The Role of PD-L1 in the Immunopathogenesis of Chronic Gastrointestinal Diseases

As discussed above, signaling through PD-L1 is critical to gut immune homeostasis, and its abnormality results in acute inflammation in animal models [42,52,53]. Thus, it is logical to assume that abnormal PD-L1 expression and/or signaling is involved in the gut chronic inflammatory diseases such as Crohn’s disease (CD), ulcerative colitis (UC), celiac disease, as well as chronic infections such as *Helicobacter pylori*. Furthermore, taking into consideration the critical role of PD-L1 in the mucosal tolerance and cancer, it seems likely that abnormality in this signal may prevent the resolution of inflammatory responses contributing to the development/persistence of chronic inflammation. Because the role of this molecule in *Helicobacter pylori* infection was reviewed by us and others previously [54,55,56], herein we will focus on the discussion of the contribution of PD-L1 to the immunopathogenesis of CD and UC, as well as review current knowledge on its role in celiac disease.

### 4.1. Alterations in the Expression of PD-L1 in Crohn’s Disease (CD) and Ulcerative Colitis (UC): Contribution to Chronic Inflammation

Aberrant expression of PD-L1 by innate and adaptive immune cells has been reported in both major types of IBD, UC and CD [8,42,53]. However, some contradiction in the pattern of expression of this molecule between CD and UC remains. The upregulation of PD-L1 mRNA and total protein in UC was reported by us [12] and others [38,57]. There are conflicting reports in the field, where one report shows that while expression may not be changed in ileal disease and two studies confirmed the upregulation of PD-L1 expression in inflammation CD tissue [13]. Aberrant signaling regulating PD-L1 expression in response to microbial ligands in mucosal professional antigen presenting cells (APCs) and peripheral blood monocytes in CD was reported [13]. The finding that PD-L1 expression is high on these APCs in UC colitis was noted [9,13,58]. Earlier, Kanai et al. observed that the number of PD-L1-expressing lamina propria professional immune cells was higher in both types of IBD. A similar observation was recently published by Cassol et al. [10]. Interestingly, Faleiro et al. also reported that the number of PD-L1-expressing DCs was decreased in ileal mucosa and peripheral blood of patients with active CD, while the intensity of PD-L1 expression per cell on DCs was upregulated [9].

A moderate increase in PD-L1 expression by intestinal epithelial cells in both types of IBD was seen by us [12] and others [8]. Cassol et al. found similarities in the upregulation of PD-L1 expression by epithelial cells between IBD and colitis induced by immune checkpoint therapy in cancer patients, suggesting some similarities in the pathogenesis between those diseases [10]. We observed that in contrast to a moderate upregulation of PD-L1 in both CD and UC colonic epithelium and other lamina propria cells of non-mesenchymal origin, only mesenchymal cells showed a differential change in the expression of PD-L1 between these two subtypes of IBD [12]. Thus, it is likely that changes in the PD-L1-mediated immune regulatory function of mesenchymal stromal cells may be functionally specific to the IBD subtype.

Several studies have reported that T cells play a key effector role in both CD and UC pathogenesis [59]. Naïve CD4^+^ T cells are activated by antigen-specific signals from APCs, and influenced by cytokines to differentiate into effector T-helper cells: Th1, Th2, Th17, Th9 cells, or Th22 and Tregs [60,61]. Inappropriate activation and maintenance of inflammatory responses and the lack of control through abnormality in the regulatory mechanisms have been suggested to lead to chronic intestinal inflammation [60]. Thus, it is likely that abnormality in PD-L1 signaling, as well as its expression, may be involved in the dysregulation of at least T cell responses in IBD. However, there are only a few publications on the implication of changes in PD-L1 impacting the dysregulation of T cell responses in IBD. We demonstrated that an increase in the PD-L1 expression on UC- and a decrease on CD-derived mesenchymal cells resulted in the respective changes of these cells capacity to suppress Th1 type responses; decreased suppression of Th1 in CD, with enhanced suppression in UC [12]. Furthermore, a disruption of Treg function has been observed in the IBD mucosa, which may also be attributed to PD-L1 [62,63]. In our previous study, we showed that while mesenchymal cells isolated from normal colonic mucosa induce the generation of the suppressive Tregs, in IBD these cells lose their capacity to generate fully suppressive Tregs [64]. Our more recent work suggests that the decrease in surface PD-L1 on these cells contributes to increasing in Th17 immune responses in CD [37]. Further studies are needed to understand the contribution of the changes in PD-L1 signaling between innate immune and adaptive immune cells in the immunopathogenesis of IBD.

While several animal studies outlined the importance of PD-L1/PD-1 signaling in the gastrointestinal pathobiology (Table 1), reports on the role of PD-L1 and its receptor PD-1 in murine models of chronic colitis mimicking IBD remain contradictory. PD-1 deficiency impairs the induction of regulatory T cells and promotes severe CD-like colitis [65]. PD-L1 expression by DX5^+^NKT cells induces apoptosis of colitogenic CD62L^+^CD4^+^ T cells in vitro. This may be one of the mechanisms which can protect mice in the model of chronic colitis induced by the transfer of CD45RB^high^ [66]. Suppression of PD-L1 with anti-PD-L1 monoclonal antibodies (mAbs) was reported to reduce chronic intestinal inflammation in the T cell transfer murine model of colitis in SCID mice [67]. The use of PD-L1 blocking mAb in this model prevented the development of experimental colitis in association with the decreased expansion of pathogenic T cells and downregulated inflammatory cytokine production (i.e., IFN-γ, TNF-α) by lamina propria CD4^+^ T cells [67]. By contrast, in colitis induced by dextran sulfate sodium (DSS) and T cell transfer, the addition of PD-L1 through PD-L1-Fc treatment significantly ameliorated the worst level of colitis [52,53]. The decrease in interleukin IL17-producing CD4^+^ T cells and an increase in IFN-γ-producing CD4^+^ T cells in the colon of DSS-treated mice upon treatment with PD-L1-Fc were also reported [53]. Furthermore, PD-L1-deficient mice were highly susceptible to DSS- or TNBS-colitis [42]. Knockdown of PD-L1 in DSS murine colitis resulted in significant increase of TNF-α, whereas the levels of the cytokines IL-6, IL-2p70, IL-4, IFN-γ, and MCP-1 were comparable. Interestingly, in this study, PD-L1 expression by epithelial and mesenchymal stromal cells, but not hematopoietic cells, was found to be critical to the control over the above inflammatory responses [42].

Contradictory results with the use of PD-L1 mAbs in animal models may be at least partially due to a direct cytotoxic effect of these antibodies on T cells, since activated T cells were reported to express PD-L1 [70,71]. This is supported by the reports obtained in an animal model of alloreactivity graft versus host disease (GvHD), where PD-L1^−/−^ T cells promoted less gut injury in recipients than the WT T cells [69]. In another study, PD-L1 Ig treatment significantly suppressed T cell proliferation, promoted T cell apoptosis, and reduced pro-inflammatory cytokine expression by effector T cells in vitro [71]. Additionally, some differences in the results pertinent to the role of PD-L1-mediated signaling in IBD may be due to the limitation of the used animal model, as well as the timing of the PD-L1 blockade: initiation versus chronicity of the IBD mimicking inflammation. Ultimately, additional preclinical studies are required to delineate the mechanism by which change in this molecule expression on a given subset of mucosal cells contributes to the development and progression of CD and UC.

### 4.2. The Role of PD-L1 in IBD Associated Fibrosis

Intestinal fibrosis develops as a result of chronic inflammation in IBD, especially in CD. Up to 50% of CD patients have clinically apparent intestinal obstructions due to fibrostenosis over their lifetime. At the same time, between approximately 1% and 11% of UC patients develop colonic stenosis [72]. No curative treatment is currently available for fibrotic complications in IBD. Increase in mesenchymal cell proliferation and deposition of extracellular matrix (ECM) characterize the pathophysiology of intestinal fibrosis. While the mechanisms inducing this abnormal activation of mesenchymal cells is far from understood, it is clear that crosstalk between these cells and professional immune cells during IBD progression is a key to this process.

PD-L1 may also play a role in fibrosis. While not yet analyzed in IBD, in an animal model of lung fibrosis, the upregulation of PD-L1 was shown on invasive lung fibroblasts [73]. Moreover, PD-L1 was essential for the lung fibroblasts having an invasive phenotype and implicated p53 and FAK-mediated signaling. Blocking FAK signaling resulted in downregulation of PD-L1, while the use of anti-PD-L1 blocking mAbs inhibited fibroblast migration and invasion, as well as lung fibrosis in humanized SCID mouse model of pulmonary fibrosis [73]. Blocking of PD-L1/PD-1 by using PD-1^−/−^ mice or PD-L1 blocking mAbs in a mouse bleomycin model of fibrosis reduced fibrosis symptoms [74]. In contrast in another humanized mouse model, administration of human MSC was shown to require PD-L1 signals to alleviate bleomycin-induced fibrosis [75]. While further studies are needed to understand the role of PD-L1 signaling in IBD-associated intestinal fibrosis, data from lung fibrosis support the idea that PD-L1 can contribute to this process.

### 4.3. Regulation of PD-L1 Expression: Implication in IBD

While little is known about the mechanisms implicated in the PD-L1 dysregulation in IBD, over the last decade, significant knowledge on the regulation of this immune checkpoint in cancer was gained due to its use as a target for cancer therapy. It has been shown in cancer cells that PD-L1 can be regulated at the levels of genomic amplification, epigenetic regulation, transcriptional regulation, posttranscriptional regulation, translational regulation, and posttranslational modification [76]. The signaling pathways of PD-L1 regulation were studied in detail for different types of cells and discussed in several reviews [76,77,78]. However, the relative contribution of the regulatory pathway to PD-L1 expression may depend on the cell type and type of stimuli and may vary depending on the type of pathogenesis [79]. Verification of PD-L1 molecular regulatory mechanisms in specific tissue types in the particular pathological process is needed.

PD-L1 is inducible on a variety of cell types, particularly on professional and non-professional APCs [80] granulocytes [81,82], lymphocytes, and, also, on different types of malignant cells [83]. Upregulation of PD-L1 can be induced by pro-inflammatory mediators such as IFN-γ, TNF-α, and IL-17A [46,47,48,49]. The role of these cytokines in the immunopathogenesis of IBD is well studied [25,84,85]. Perhaps the most studied mechanism of PD-L1 upregulation is through IFN-γ and shown to involve JAK/STAT1 signaling pathway, as well as activation of NF-kB transcription factor and transient phosphorylation of ERK1/2 and PI3K [86,87]. Importantly, activation of the Akt/mTOR pathway tightly regulates PD-L1 expression in vitro and in vivo [79]. Several studies are shown to upregulate PD-L1 through the PI3K/Akt/mTOR signaling pathway [79], the increased activity of which was found in CD tissue [88,89]. Activation of the PI3K/Akt/mTOR pathway caused by the suppression of PTEN may be involved in this pathogenesis. PTEN is involved in the regulation of intestinal permeability and is also defined as a tumor suppressor molecule of the PI3/Akt pathway in the inflammatory response, cell migration, and proliferation [88]. PTEN mRNA and protein expression in peripheral CD4^+^ T cells or mucosal lymphocytes was lower in CD patients than in healthy controls [89]. PTEN expression was observed to be decreased in the inflamed colonic mucosa of IBD compared to the non-inflamed mucosa [88].

The IBD-relevant regulatory factors, IL-10 [49] and TGF-β [52], are also known to upregulate PD-L1 on APCs. The blockade of TGF-β1 downregulated PD-L1 expression and precipitated graft rejection in the model of pancreatic islet transplantation [90]. However, the data on the direct effect of TGF-β1 on PD-L1 remain controversial. It was established that the treatment of renal proximal tubular epithelial cells or blood monocytes with TGF-β did not impact PD-L1 expression [52,91]. By contrast, TGF-β1 was found to upregulate PD-L1 gene transcription in a SMAD2-dependent manner, and a positive association between PD-L1 and phosphorylated SMAD2 was found in NSCLC tumors [92]. TGF-β has been suggested to increase PD-L1 expression in tumor cells through epigenetic remodeling and posttranslational modifications (in ref [93]). Recently, Garo et al. demonstrated SMAD-dependent PD-L1 regulation by TGF-β in a colitis model relevant to IBD, where signaling through SMAD2 was critical to increased PD-L1 expression, while the activation of SMAD7, an inhibitor of TGF-β canonical signaling pathway, downregulated PD-L1 [52]. Interestingly, the upregulation of SMAD7 in the colon in patients with active IBD was shown [51,94,95].

Stimulation of pattern recognition receptors (PRR) with microbial ligands was also reported to increase PD-L1 expression in vitro [48,58,96]. Internalization of peptidoglycan in myeloid cells was reported to induce PD-L1 expression by Myd88-independent, but RIP2-dependent process [97]. The role of NOD2 gene variants in CD patients in cellular responses to muramyl dipeptide (MDP) has also been shown to affect PD-L1 expression. Myeloid cells from CD patients homozygous for *NOD2* L1007fsincC variant failed to induce PD-L1 in response to stimulation with MDP, while CD patients with the heterozygous genotype still responded to MDP with increased PD-L1 expression [58]. Signaling through MyD88-dependent TLR4 and TLR5 were also shown to be involved in the immunopathogenesis of IBD [36,98]. The impact of IBD associated genetic polymorphisms in these PRRs on the regulation of PD-L1 expression remains to be determined.

## 5. Celiac Disease

Celiac disease is a chronic inflammatory disorder that results from a loss of gluten tolerance. Pathogenesis of celiac disease involves an increase in activated CD4^+^ T cells that recognize deamidated gluten peptides in HLA-DQ complexes [99]. Increased production of IFN-γ by activated T cells causes B cells to produce disease-specific autoantibodies that recognize the enzyme transglutaminase 2 (TG2) and activate CD8^+^ T cells to lyse epithelial cells [16]. While extremely limited information is currently available on the role of PD-L1-mediated signaling in celiac disease, a recent study suggests that alteration of the PD-L1/PD-1 pathway is involved in the immunopathogenesis of this disease [16]. Increased serum levels of soluble PD-1 (sPD-1) and sPD-L1 were found in patients with active disease when compared to healthy controls. Furthermore, increases in surface PD-L1 expression on intestinal epithelial and lamina propria cells of these patients was also reported. Excessive sPD-1 found in serum of these patients was suggested to compete with surface-associated PD-1, blocking the PD-L1/PD-1-mediated suppression of T cell responses and resulting in an aberrant increase of T cell proliferation [16]. Despite this information, further studies are need to clarify the role of this pathway in celiac disease.

Interestingly, an overlap (up to 10%) of patients with CD (but not UC) and celiac disease suggests there may be a genetic link between the two [100]. The genes shared between CD and celiac disease are involved in innate immune response against pathogens and in the adaptive immune system activation, in particular Th17 cells [101]. Remarkably, we recently reported that increase in s-PD-L1 and decrease in surface-associated PD-L1 on mesenchymal component of intestinal mucosa might contribute to the increase in Th17 immune responses in CD [37]. This further suggests that the potentially overall role of PD-L1 in the immunopathogenesis of both CD and celiac disease should be investigated in regards to the PD-L1-mediated regulation of Th17 cells.

## 6. PD-L1/PD-1 as a Potential Therapeutic Target in Gut Chronic Inflammatory Diseases: Lesson Learned from Immune Checkpoint Therapy of Solid Cancers

PD-L1 expression on tumor cells and PD-L1 and PD-1 expression on immune cells are key mechanisms by which tumor cells escape anti-tumor immune surveillance via the suppression of the anti-tumor effector T cell response [22]. PD-L1 expression is significantly increased in several solid tumors, including microsatellite instability (MSI) colon cancer [102,103]. At the same time, in some colorectal cancers, PD-L1 is only expressed on tumor-infiltrating cells and is rarely found on tumor cells [103]. This places great significance on the role of PD-L1 in the tumor microenvironment. Indeed, stromal PD-L1 was associated with less aggressive tumor progression in colon cancer patients and better survival [18].

Nowadays, immune checkpoint inhibitors have emerged as a remarkable treatment option for diverse cancer types. However, a significant number of patients on checkpoint inhibitors develop immune-related adverse events (irAEs), affecting a wide variety of organs [104]. The most common adverse events of checkpoint blockade are gastrointestinal-adverse events [105,106]. Immune checkpoint-induced colitis (ICI colitis) is considered a distinct form of colitis with an acute onset and rapid progression, which leads to potential complications, including bowel perforation [107]. This type of side effect usually develops within the first few weeks or months after start of treatment but can occur at any time, including after stopping immune checkpoint blockade therapy [108].

A dominant feature of ICI-active colitis is the neutrophilic infiltration, crypt microabscesses, and prominent crypt epithelial cell apoptosis. However, a lymphocytic colitis-like pattern with increased intraepithelial lymphocytes was also observed [107,109]. Remarkably, PD-L1/PD-1 expression depended on ICI colitis recurring as a chronic histological manifestation consisting of basal lymphoplasmacytosis, crypt architectural irregularity, and a case of Paneth cell metaplasia, all reminiscent of IBD [110]. Also similar to IBD, ICI colitis is characterized by immunological changes, such as (1) CD4^+^ T cells predominant mucosal lamina propria infiltrate, and Th1/Th17 upregulation with normal Th2 expression (observed as well in CD); (2) elevated expression TNF-α and TNFR-like proteins (observed in both, CD and UC); and (3) mucosal abnormality in the expression of Foxp3 and IL-10 (observed in both, CD and UC) [105].

Recent data demonstrated the critical role of microbiota in predicting the development of ICI colitis and impacting the expression of PD-L1 and/or its receptor PD-1 [105]. While further studies are required to evaluate the safety of the use of probiotics in cancer patients, the use of probiotics may be helpful to render tumors sensitive to the PD-L1/PD-1 immune checkpoint therapy through the modulation of expression of these molecules. However, there is a gap in our knowledge when it comes to understanding the impact of gut microbiota on PD-L1 expression and regulation of PD-L1/PD-1 signaling. However, recent studies demonstrated that the commensal enteric strain of *Escherichia coli* K12 upregulated PD-L1 in IFN-γ sensitized colonic cancer epithelial cell lines, and that this process was NF-κB-dependent [111]. Interestingly, the same study showed that *E. coli* strain L20, which was isolated from a CD patient, did not affect PD-L1 expression. Furthermore, another commensal strain, *Enterococcus faecalis,* even significantly inhibited PD-L1 expression [111]. Other groups recently demonstrated that commensal strain of *Pediococcus pentosaceus* sp. derived extracellular membrane vesicles upregulated PD-L1 on bone marrow-derived macrophages and bone marrow progenitor cells in culture and induced recruitment of PD-L1-expressing myeloid cells to the wound site in vivo [112]. Furthermore, the probiotic strain of *Bifidobacterium infantis* has been shown to upregulate PD-L1 in a model of acute murine colitis [113]. Thus, a further understanding of the interactions between the gut microbiome and PD-1 ligands is needed as a potential avenue to modulate these immune checkpoints in IBD and ICI.

Patients with autoimmune disease or immune-related predisposition, such as IBD, have been excluded from most immune checkpoint clinical trials because of the potential for increased toxicity [114]. Indeed, gastrointestinal adverse events, such as diarrhea, in patients with underlying IBD who received immune checkpoint inhibitors, are at high risk [115]. Notwithstanding, treatment with anti-PD-1 antibodies in melanoma patients induce relatively frequent immune toxicities in patients with baseline autoimmunity/chronic inflammatory diseases or prior immune-related adverse events. However, recurrence of ICI colitis was rare even in patients with preexisting autoimmune diseases that were treated with TNF-α inhibitors [68]. Therefore, the benefits versus risks of the PD-L1/PD-1 immunotherapy in this population should be considered [68,114]. Together, these data suggest that preexisting autoimmune conditions are not an absolute contraindication to the immune checkpoint inhibitors therapy, but need a thoughtful approach. Thus, lessons learned from immune checkpoint therapy as well IBD preclinical models suggest that the effectiveness of at least PD-L1 blockade or its “supplementation” may be considered as a therapeutic approach in patients with chronic inflammatory disease of the gut where abnormality in PD-L1/PD-1 signals were reported. Therefore, understanding the potential of these molecules as a therapeutic target in several types of IBD and celiac disease warrants further investigation.

## 7. Conclusions

While dramatic progress has been made in the understanding of the role of PD-L1, in the maintenance of gut homeostasis and inflammatory disease, it is far from complete. It is clearly established that PD-L1 is critical to gut mucosal tolerance. Abnormality in PD-L1 expression and/or signaling was observed in gut chronic inflammatory diseases such as Crohn’s disease (CD), ulcerative colitis (UC), celiac disease, as well chronic infections such as *Helicobacter pylori* (Figure 1). However, how and why these abnormalities occur and their impact on disease development and progression requires further investigation. Furthermore, a more extensive understanding of the PD-L1 intrinsic signaling in cells implicated with the discussed GI pathologies is warranted. The impact of the membrane associated versus soluble form of PD-L1 in gut health and diseases warrant further investigation. With this consideration, the role of commensal and dysbiotic microbiota in the regulation of these immune checkpoints should be further investigated. Finally, the mechanisms contributing to the regulation of PD-L1 expression and its interaction with other positive and negative B7 co-stimulators during gut homeostasis and chronic inflammatory diseases are only at the beginning of understanding. Therefore, further studies are needed to address these key gaps in the field and provide a scientific basis for the design of novel PD-L1 targeting approaches for the treatment of chronic inflammatory disease of the gut.

## Figures and Tables

**Figure 1 ijms-21-09165-f001:**
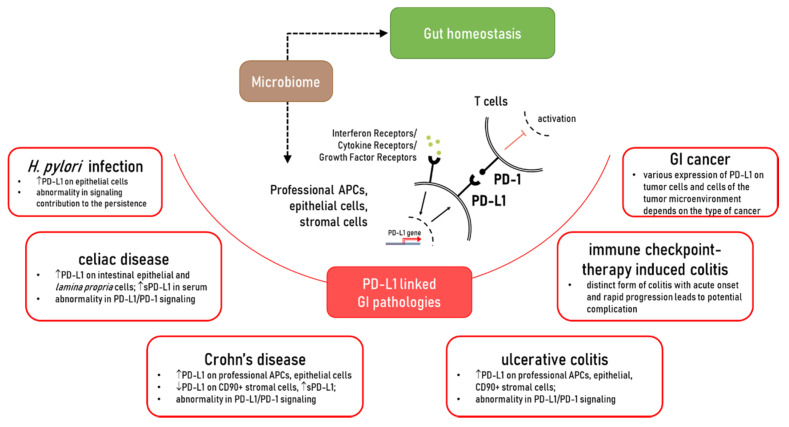
The role of PD-L1 in the gut homeostasis and pathologies which affected PD-L1 expression and/or PD-L1/PD-1 signaling.

**Table 1 ijms-21-09165-t001:** Summaries of studies on the role of PD1/PD-L1 in gastrointestinal pathology.

What Was Shown	Animal Model	PD-L1 Source and Regime	Mechanism of Action	Ref.
PD-L1-mediated protection	DSS- and TNBS-colitis in PD-L1^−/−^; Rag-1^−/−^; PD-1^−/−^ C57BL/6 background mice	Knockout	PD-L1-mediated protection during intestinal inflammation through epithelial cells or myofibroblasts and pericytes. Adaptive immune response was not required for PD-L1 protection in this model.	[42]
PD-L1-Fc displayed protective effects	DSS-colitis in wild-type B6	Adenovirus expressing PD-L1-Fc (Ad/PD-L1-Fc) at a dose of 1 × 10^8^ plaque forming units (PFU) on days 0, 3, and 6, or PD-L1-Fc protein at a dose of 30 μg on day 2	PD-L1-Fc displayed protective effects on both systemic and mucosal inflammation, due to the downregulation of IFN-γ and IL-17 production. PD-L1-Fc might exert a protective effect by targeting different types of immune cells driving inflammation, such as myeloid cells in the DSS model and CD4 Th cells in the T-cell transfer model.	[53]
	T-cell-induced colitis model in Rag-1 KO mice	Three weeks after T-cell reconstitution, recombinant PD-L1-Fc protein was administered at 1-week intervals at a dose of 20 μg for 4 weeks		
PD-L1-Fc displayed protective effects	DSS-colitis in C57BL/6J WT	PDL1-Fc treatment were given i.p. (100 mg per mouse every other day, four times)	This colitis mitigation in PDL2-Fc- or PDL1-Fc-treated mice is associated with increased colonic Foxp3^+^T cells.	[52]
Blocking PD-L1 is reduced ability of DX5^+^NKT to killing of colitogenic T cells	Chronic DSS-colitis in BALB/c SCID mice	Anti-PD-L1 antibody	In vitro blocking PD-L1 with specific antibody resulted in a reduced ability of DX5^+^NKT cells to induce the death of CD62L^+^CD4^+^ T cells and lymphocytes derived from intestinal lymph node tissue of mice with chronic DSS-mediated colitis.	[66]
Anti-PD-L1 mAb prevents the development of colitis	Adoptive transfer of CD4^+^CD45RB^high^ T cells to SCID mice	Anti-PD-L1 (MIH6) mAb 250 μg/mouse three times per week starting on the day of T cell transfer and continuing up to 7 weeks	Decreased expansion of pathogenic T cells and the down-regulated Th1 cytokine production (i.e., IFN-γ, IL-2, and TNF-α) by LP CD4^+^ T cells. The endogenous PD-L1 might be required for the expansion and differentiation of adoptively transferred CD4^+^CD45B^high^ T cells in vivo.	[67]
PD-1^−/−^ mice are resistant to colitis	DSS-induced colitis in specific-pathogen-free PD-1^−/−^ mice in C57BL/6 (B6)-background	PD-1 KO mice	PD-1^−/−^ mice are resistant to DSS-induced colitis; altered microbiota in PD-1^−/−^ mice modulates gut inflammation, colon microbiota of PD-1^−/−^ mice is less colitogenic than WT commensal bacteria.	[68]
PD-1 expression on CD4^+^ LP T cells associated with protection from colitis	Adoptive transfers of the colitic LP CD4^+^ T cells after developing CD4^+^CD45RB^high^ T cell in Balb/c or colitic LP CD4^+^ T cell-transferred colitis in SCID mice		LP CD4^+^ T cells obtained from non-colitic mice after over seven or more transfers expressed significantly higher levels of PD-1.	[65]
PD-L1 blockade leads autoimmune enteritis	Naive congenic CD45.1^+^ OT-I T cells were transferred i.v. into CD45.2^+^ C57BL/6 or iFABP-tOVA recipients	Mice were injected i.p. with 200 μg of anti-PD-L1 Ab on day –1, day 2, and every other day	Ab-mediated PD-L1 blockade leads to a considerable expansion of OVA-specific CD8^+^ T cells and their differentiation into effector cells capable of producing pro-inflammatory cytokines. It breaks the intestinal tolerance in iFABP-tOVA mice leading to CD8^+^T cell-mediated autoimmune enteritis.	[45]
decreased gut homing and cytokine production by Pd-L1^–/–^ donor T cells in animal model of GVHD	Lethally irradiated BALB/c recipients were infused with 10^7^ WT B6 BM cells alone or with 2 × 10^6^ WT B6 or Pd-L1^–/–^ purified T cells	PD-L1 KO donor	Upregulation of PD-L1 on GVHD-causing CD4 and CD8^+^ T cells is necessary for their survival, proliferation, and optimal T eff function. PD-L1–deficient T cells had reduced expression of gut homing receptors, diminished production of inflammatory cytokines, and enhanced rates of apoptosis.	[69]

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
