# Peer review of "Role of PD-L1 in Gut Mucosa Tolerance and Chronic Inflammation"

_ijms, 2020, doi:10.3390/ijms21239165_

Round 1
Reviewer 1 Report
The review by Chulkina et al provides a comprehensive summary on the role of PD-L1 in the homeostasis and inflammation of the gut mucosa.
The authors provide a thoroughly summary of the current knowledge in this area, with a main focus of PD-L1 in health and on chronic intestinal conditions such as Inflammatory bowel disease (IBD) and celiac disease. The review is very well written, very detailed (specially the summary table) and highlight most of the main findings reported to date in this area.
Comments
- Revise grammar and some misspellings.
- A role of the microbiota in regulating PD-L1 in the gut should be included such as recent publications Lee et al, 2020, DOI: 10.1128/IAI.00618-20; Bulut et al, 2020, DOI: 10.4049/jimmunol.2000731; Zhou et al, 2019, DOI: 10.5009/gnl18316.
- A role for the microbiota in regulating PD-L1 could be added as part of a future area to investigate under the conclusion section.
- A summary figure depicting the role of PD-L1 in the gut should be included too.
- Revise abbreviations e.g. DC, ILCs, GALT etc, the first time they’re mentioned.
- In Table 1 – as part of “PD-1-/- are resistant to colitis” there should be commas between the sentences “PD-1−/− mice are resistant to DSS-induced colitis altered microbiota in PD-1−/− mice modulates gut inflammation PD-1−/− mice is less colitogenic than WT commensal bacteria”
Author Response
We thank the Reviewer for constructive comments and suggestions, which helped us to improve the quality of the article.
Question: Revise grammar and some misspellings.
Answer: According to this request, our manuscript was revised grammar and misspelling.
Question: A role of the microbiota in regulating PD-L1 in the gut should be included such as recent publications Lee et al, 2020, DOI: 10.1128/IAI.00618-20; Bulut et al, 2020, DOI: 10.4049/jimmunol.2000731; Zhou et al, 2019, DOI: 10.5009/gnl18316.
Answer: We appreciate the Reviewer’s suggestions and agree that the microbiota plays a critical role in PD-L1 regulation in the GI. We included recent publications in the review, and now they are discussed on page 10, lines 379-396.
Question: A role for the microbiota in regulating PD-L1 could be added as part of a future area to investigate under the conclusion section.
Answer: We add this under the conclusion section, page 10, lines 422- page 11, 431.
Question: A summary figure depicting the role of PD-L1 in the gut should be included too.
Answer: We appreciate the Reviewer’s suggestions, and added the summary figure describing the role of PD-L1 in the gut on page 11.
Question: Revise abbreviations e.g. DC, ILCs, GALT etc, the first time they’re mentioned.
Answer: We spell out all abbreviations in the text as requested.
Question: In Table 1 – as part of “PD-1-/- are resistant to colitis” there should be commas between the sentences “PD-1−/− mice are resistant to DSS-induced colitis altered microbiota in PD-1−/− mice modulates gut inflammation PD-1−/− mice is less colitogenic than WT commensal bacteria”
Answer: We correct this, see table 1 on page 6 as described below:
“PD-1−/− mice are resistant to DSS-induced colitis; altered microbiota in PD-1−/− mice modulates gut inflammation, colon microbiota of PD-1−/− mice is less colitogenic than WT commensal bacteria.”
Reviewer 2 Report
This is a very nice review on PDL1 but instead of focusing it in cancer, the authors have concentrated on the gut and its implication in other disorders apart from cancer.
It is very well written,easy to read, with very useful information for the reader. I believe the paper could be published just as it is.
Nevertheless, if the authors would be interested in giving it another novel twist, it is known that PD-L1 drives intracellular signaling into cancer cells and T cells, interfering with IFN-driven apoptosis but also affecting T cell activities by modulating intracellular signaling pathways. The authors may be willing to link some of PD-L1 intrinsic signaling into cells with pathology in the GI tract, for example.
Author Response
Comments and Suggestions for Authors
This is a very nice review on PDL1 but instead of focusing it in cancer, the authors have concentrated on the gut and its implication in other disorders apart from cancer.
It is very well written, easy to read, with very useful information for the reader. I believe the paper could be published just as it is.
Nevertheless, if the authors would be interested in giving it another novel twist, it is known that PD-L1 drives intracellular signaling into cancer cells and T cells, interfering with IFN-driven apoptosis but also affecting T cell activities by modulating intracellular signaling pathways. The authors may be willing to link some of PD-L1 intrinsic signaling into cells with pathology in the GI tract, for example.
Answer: We very much appreciate the Reviewer’s found our review informative and well written. While we agree with the Reviewer on the importance of the PD-L1 signaling in cancer cells, this topic was thoughtfully reviewed in several recent publications including but not limited: Sun et al., 2018; Zerders et al., 2018; Ju et al., 2020. Thus, in a current review, we choose to focus on the role of PD-L1 in the IBD as the less covered and less discussed are. We also agreed with the Reviewer that PD-L1 intrinsic signaling is the topic, which needs to be studied and discussed. While very limited knowledge is currently available on the PD-L1 intrinsic signaling within different cell subsets in GI pathologies, we incorporate what discussion of current knowledge in this area under section 4.8, on page 8, lines 283-295, 304-308, and 310-327. Considering this reviewer suggestion, we also include under the conclusion section sentence about the importance of future studies in this area.
Reference cited in the response for Reviewer:
- Sun, C.; Mezzadra, R.; Schumacher, T.N. Regulation and Function of the PD-L1 Checkpoint. Immunity 2018, 48, 434–452.
- Zerdes, I.; Matikas, A.; Bergh, J.; Rassidakis, G.Z.; Foukakis, T. Genetic, transcriptional and post-translational regulation of the programmed death protein ligand 1 in cancer: biology and clinical correlations. Oncogene 2018, 37, 4639–4661.
- Ju, X.; Zhang, H.; Zhou, Z.; Wang, Q. Regulation of PD-L1 expression in cancer and clinical implications in immunotherapy. Am. J. Cancer Res. 2020, 10, 1–11.